# Investigation of the Combined Effect of Total Ionizing Dose and Time-Dependent Dielectric Breakdown on PDSOI Devices

**DOI:** 10.3390/mi13091432

**Published:** 2022-08-30

**Authors:** Jianye Yang, Hongxia Liu, Kun Yang

**Affiliations:** Key Laboratory for Wide Band Gap Semiconductor Materials and Devices of Education, School of Microelectronics, Xidian University, Xi’an 710071, China

**Keywords:** time-dependent dielectric breakdown, total ionizing dose, threshold voltage, gate oxide

## Abstract

The combined effect of total ionization dose (TID) and time-dependent dielectric breakdown (TDDB) of partially depleted silicon-on-insulator (PDSOI) NMOSFET is investigated. First, the effect of TDDB on the parameter degradation of the devices was investigated by accelerated stress tests. It is found that TDDB stress leads to a decrease in off-state current, a positive drift in threshold voltage, and a reduction of maximum transconductance. Next, the degradation patterns of TID effect on the devices are obtained. The results show that the parameter degradation due to gamma radiation is opposite to the TDDB stress. Finally, the combined effect of TID and TDDB is investigated. It is found that the drift of the devices’ sensitive parameters due to TDDB stress decreases in a total dose of gamma radiation environment. The TDDB lifetime is shortened, but the pattern of gate current change remains unchanged. The failure mechanism of the gate oxide layer under TDDB stress is not changed after irradiation.

## 1. Introduction

Electronic devices in aerospace applications are subjected to both space irradiation and electrical stress as they operate in a space-irradiated environment for long periods. Silicon-on-Insulator (SOI) technology has been widely used because of its main advantages. SOI technology eliminates the latch-up effect, and reduces soft error rates and parasitic capacitance compared to bulk silicon devices. However, the unique device structure of the SOI technology results in a different irradiation damage effect than that of bulk silicon devices in harsh space radiation environments.

It has been shown that the space irradiation environment can lead to the generation of hole traps in the gate oxide layer of MOS devices and interface traps at the Si-SiO_2_ interface, causing the total dose effect and single particle effect, leading to degradation of the sensitive parameters of the device and even causing circuit failure [1,2,3]. SOI devices achieve complete dielectric isolation through the buried oxygen layer (BOX), which has a significant advantage in resisting the single-event-transient effect. However, BOX and shallow trench isolation (STI) cause SOI devices to be more significantly affected by TID effects [4,5,6]. In addition, the irradiation effect couples with the reliability effects such as channel hot carrier injection (HCI) and negative bias temperature instability (NBTI), resulting in changes in the degree of device damage [7,8]. Some researchers experimentally found that different types of irradiation have different effects on the TDDB lifetime of PDSOI. When subjected to gamma irradiation, the generation of oxide traps reduces the device FN tunneling voltage, resulting in a lower TDDB lifetime, whereas proton irradiation leads to an increase in the TDDB lifetime of PDSOI devices on P-type substrate [9,10]. Ying W. investigated the effect of gamma irradiation on TDDB lifetime using PMOS capacitive structures produced on a 65 nm process. It is found that the change in the gate current profile (Ig-Vg) of the device before and after irradiation is minimal, whereas the TDDB lifetime of the gate capacitor was significantly reduced [11]. For the reliability of gate dielectric irradiation of deep submicron SOI MOSFET devices, most of the existing studies have been conducted by experimentally obtaining sample data before and after irradiation, based on which a mechanistic analysis is performed, with less attention paid to the changes in critical parameters during irradiation and voltage stress, such as threshold voltage, subthreshold characteristics, gate current, and transconductance. To ensure that electronic devices can operate stably in the space environment for a long period of time, it is essential to study the changes in the sensitive parameters of the devices during multi-stress events.

In this paper, PDSOI NMOSFETs are used as experimental objects. Different experiments are set up to test the TDDB lifetime of the PDSOI devices before and after irradiation as well as the transfer characteristic curves, threshold voltage, drift of the trans-conductance, and TDDB lifetime during the stressing process. The combined effect of TDDB and TID on the PDSOI devices is investigated.

## 2. Materials and Methods

All devices used in the experiments were fabricated based on the PDSOI process. The devices have a channel area of 10 μm × 1.2 μm, which is less affected by parasitic effects. Base wafers are from Shanghai Huahong Grace Semiconductor Manufacturing Corporation. The layout is shown in Figure 1. The T-gate structure is implemented with body contact to suppress the floating-body effect. The top silicon layer thickness is 100 nm, the buried oxygen layer is 145 nm thick, the gate oxide layer is 7 nm thick, and the operating voltage is 3.3 V. DIP packages are used to improve test reliability.

To investigate the total ionizing effects on time-dependent dielectric break-down characteristics, the samples were divided into two groups. One group is used for the TDDB accelerated stress experiments only and the other is for the gamma irradiation experiments followed by the TDDB accelerated stress experiments, which were carried out using the constant voltage method. Each sample was initially measured before stress was applied to prevent damaged samples from affecting the experimental results. When the gate current Ig is greater than ten times the initial value during the experiment, the sample is judged to have broken down, and the experiment is stopped. The irradiation experiments were carried out using a Co60 gamma irradiation source provided by the Xinjiang Technical Institute of Physics and Chemistry. The temperature was kept at 25 °C, and all devices were biased at ON state (Vg = 3.3 V, Vd = vs. = Vb = 0 V) during the experiments. The dose test points were 100 krad (Si), 200 krad (Si), 300 krad (Si), 400 krad (Si), 500 krad (Si), and 1000 krad (Si), with a pause at each set dose point to perform a parametric test of the device and to complete it within 10 min.

The electrical parameters of the devices were measured and extracted using an automated semiconductor parameter measurement system consisting of a computer and a B1500A [12] semiconductor parameter analyzer. Breakdown of the devices was detected in time, and test errors were minimized. A pre-programmed test program was used to extract sensitive parameters such as transfer characteristic curves, transconductance, and the device’s threshold voltage during accelerated stress, as well as to determine the device status and automatically stop the test when the conditions were met.

## 3. Results and Discussion

In this paper, three parts of the experiment will be analyzed. Firstly, the data from a single TDDB stress experiment will be analyzed to obtain the degradation pattern of the sensitive parameters of the PDSOI device when subjected to TDDB stress. Secondly, the sensitive parameters of the device at each dose point will be extracted from the results of the TID experiment, and the effect of irradiation on the sensitive parameters of the device will be compared and analyzed. Finally, the effect of TDDB stress on the sensitive parameters of the experimental samples will be analyzed before and after the TID irradiation experiment. The combined effect of TDDB and TID was obtained.

### 3.1. Effect of TDDB Stress on Sensitive Parameters of PDSOI Devices

The TDDB process is generally considered to be divided into two phases. The first phase is the defect accumulation phase when a large number of traps are generated and accumulated within the gate oxide layer as the stress time increases. The second phase is the breakdown phase. When the trap density accumulated in the first phase reaches a critical value, a complete conductive path will be formed inside the gate oxide layer. At this point, the gate current will rise rapidly along with the electrical and thermal positive feedback effects, eventually leading to a breakdown of the gate oxide layer. This experiment focuses on the first phase of the TDDB process. The degradation of output characteristic curve, threshold voltage, and transconductance due to the accumulation of traps during accelerated stress were observed. Figure 2 shows the degradation of the TDDB first phase transfer characteristic curves for gate voltages Vg = 7 V (a) and Vg = 7.5 V (b) at 25 °C. There is a slight change in the device’s off-state current as the stress time increases. It can be observed that the positive shift of the curve and the subthreshold characteristics become worse. As the gate stress voltage increases to 7.5 V, the device transfer characteristic curve rapidly shifts positively for the first 1000 s as the TDDB stress time increases. The drift then decreases and tends towards saturation. The threshold voltage represents the device channel opening voltage and is closely related to the oxide layer charge and the interface charge at Si/SiO_2_ [13], which can reflect the change in charge inside the device. Figure 3 gives the experimental threshold voltage variation curve with increasing stress time. When the stress voltage was 7 V, the variation of the device threshold voltage with stress time was closer to a linear relationship. As the voltage stress increases to 7.0 V or 7.5 V, the threshold voltage rose rapidly for the first 1000 s and then changed slightly into saturation. Figure 4 provides the variation of the transconductance curve of the device with stress time for the same experimental conditions. It can be observed that the TDDB stress causes the transconductance curve of the sample device to drift to the right, while the transconductance maximum decreases with increasing stress time, as shown in Figure 5. At a stress magnitude of 7.5 V, the maximum transconductance value decreased by approximately 20% compared to the initial value by 1000 s and by only 6% at 6000 s compared to 1000 s.

The main reason for the drift of the device output characteristic curve, threshold voltage, and transconductance is that the internal Si-O bond will break to form an oxide trap due to the strong electric field effect inside the gate oxide layer caused by the TDDB stress. At the same time, the positively charged hydrogen ions (H+) released from the gate under the action of the strong electric field accumulate across the oxide layer at the Si-SiO_2_ interface and react with the Si-Si bonds at the interface or activate the Si suspension bonds near the interface. This prevents the channel from opening; the output characteristic curve drifts positively, and the threshold voltage rises. The accumulation of interface traps near the channel enhances the carrier scattering effect, decreasing trans-conductance. The higher the stress voltage, the faster the traps accumulate inside the oxide layer and at the interface. The trap concentration will reach saturation when the traps accumulate to a certain number. At this time, the variation of the sensitive parameters of the device will decrease with the stress time. When the trap density reaches a critical value, the device will enter the second stage of TDDB, the breakdown phase.

### 3.2. Evaluationof the TID Effect on Sensitive Parameters of PDSOI Devices

The main mechanism of damage caused by total dose irradiation to PDSOI devices is that total dose irradiation causes positively charged oxide trap charges in the gate oxide layer and buried oxygen layer, resulting in an increase in channel carrier concentration and a decrease in channel potential barrier. At the same time, the charge in the buried oxide layer causes the formation of leakage paths induced above the buried oxide layer, which reduces the device threshold voltage. This leads to more effortless channel opening at the same gate voltage. Figure 6 shows the transfer characteristic curves and the threshold voltage of PDSOI devices under different levels of total dose irradiation. In contrast to the TDDB stress experimental results, the device off-state leakage rises by an order of magnitude, and the threshold voltage drops by about 0.1 V when the accumulated dose is 1000 krad (Si). Figure 7 gives the variation curves of the transconductance at different dose points. It can be observed that the maximum transconductance increases with increasing irradiation dose for the same experimental conditions. Notice also that the whole curve of the transconductance shifts slightly left.

The above two sets of experiments show that the effect of gamma irradiation on the sensitive parameters of PDSOI is opposite to TDDB stress. On the one hand, the threshold voltage increases with increasing TDDB stress time. The maximum transconductance decreases with increasing stress time, and the parameter drift eventually saturates. On the other hand, the threshold voltage drops, and the maximum transconductance increases as the sample devices are exposed to TID irradiation. Notice that on current is slightly improved after higher TID. The main reason is that the positive charge generated by the TID irradiation in the STI region creates a parasitic channel. However, none of the parameters drift eventually saturate in the TID environment, even when the irradiation dose reach tens of Gy [14]. The main reason for this result is the different damage mechanisms of TDDB stress and TID stress on the oxide layer of the device. The TDDB stress can produce a large number of defects in a short time, so the drift of sensitive parameters can be observed to saturate. Gamma irradiation generates electron-hole pairs within the gate oxide and buried oxygen layers through ionization. The holes drift toward the Si-SiO_2_ interface in response to the operating voltage, creating positive charge accumulation near the channel. Figure 8 presents the simulation result of electron number density in the device before and after irradiation exposure. It can be observed that the radiation leads to the channel charge accumulation. The charges are induced by the interface traps. The rate of charge accumulation from TID stress is much slower than the rate of defect accumulation from TDDB stress, so saturation does not occur.

### 3.3. Combined Effect of TID and TDDB on PDSOI Devices

Both the TID effect and the TDDB effect can cause damage to the gate dielectric of PDSOI devices. Moreover, the two have opposite effects on the sensitive parameters of the device. To investigate the combined effect of the TID effect and the TDDB effect, TDDB accelerated tests are conducted on the devices after being affected by gamma irradiation. Figure 9a shows the transfer characteristic curves when TDDB stress is applied under the same experimental conditions before and after irradiation. Similar to the above experimental results, the device’s off-state leakage current and subthreshold swing after gamma irradiation are significantly degraded. The off-state leakage current changes very little before and after TDDB stress, and the curve shifts to the right. The transfer characteristic curves of the devices with TDDB stress applied after the irradiation experiments show that the off-state current decreases slightly with increasing stress time but is still higher than the off-state current of the devices before irradiation.

Figure 9b compares the threshold voltage in the TDDB stress experiments before and after irradiation. After irradiation, there is a smaller change in the threshold voltage. The reason for this is that the oxide charge generated by gamma irradiation ionization within the oxide layer and the oxide trap charge generated by TDDB stress have opposite effects on the sensitive parameters, and the effects of the two charges cancel each other out. This eventually leads to a reduction in the drift of the sensitive parameters of the device.

The gate current curves during the final TDDB stress test for both sets of experimental samples are given in Figure 10 and represent the average of multiple sets of experimental data. The gate dielectric breakdown occurs at t = 11,015 s for the unirradiated devices and t = 7773 s for the irradiated devices. The gate current of the unirradiated device is between 10^−8^ A and 10^−7^ A during the defect accumulation phase of the stress process, and the gate current of the irradiated device is less than 10^−8^ A. The experimental results show that the gate oxide TDDB lifetime of the PDSOI device is shortened after gamma irradiation compared to that of the non-gamma irradiated device. With the increase of stress time, the traps in the oxide layer gradually accumulate, which leads to the continuous increase of the anode electric field. The oxide layer is broken down when the electric field reaches a certain critical value. As the stress time increases, the trapped electrons gradually accumulate, leading to an increasing anode electric field. When the electric field reaches a critical value, the oxide layer is broken down [15,16]. The changing pattern of gate current in TDDB accelerated tests before and after irradiation is the same, indicating that the defect accumulation and breakdown mechanism under TDDB stress is not changed after being irradiated by gamma rays. After total dose irradiation, the gate oxide, BOX, and STI layers of PDSOI devices will sense positive charges. The carrier generation near the channel increases the leakage current, and the increase in the positive charge of the oxide layer leads to a higher electron leap barrier and a lower gate current. The oxide layer charge induced by irradiation accelerates the accumulation of defects under TDDB stress and reduces the gate dielectric reliability of the device.

## 4. Conclusions

This paper analyzes and studies the combined effect of the TID and the TDDB of PDSOI devices. Multiple sets of experiments obtained the transfer characteristic curves, threshold voltage, and transconductance drift of the PDSOI devices before and after irradiation. Finally, TDDB breakdown times before and after irradiation were obtained using multiple samples. It is found that the TDDB effect is opposite to the TID effect on the sensitive parameters of the device. A single TDDB stress causes the device threshold voltage to drift positively and the maximum transconductance to decrease; the TID effect causes the device threshold voltage to drift negatively and the maximum transconductance to increase instead. In addition, TDDB stress will lead to rapid degradation and saturation of device parameters, whereas the TID effect causes lower degradation of device parameters, and the amount of degradation is linearly related to the irradiation dose. The TDDB lifetime of the device gate oxide layer is reduced after being affected by gamma irradiation. The trends of gate current during the TDDB experiments before and after irradiation are the same. This indicates that the failure mechanism of the gate oxide layer under TDDB stress is not changed after irradiation. Both stresses increase the trap concentration in the gate oxide layer, so the device gate oxygen reliability decreases, and the TDDB lifetime decreases. A further extension of our research would be to carry out material analyses such as XPS, SIMS, or XRD to confirm the results.

## Figures and Tables

**Figure 1 micromachines-13-01432-f001:**
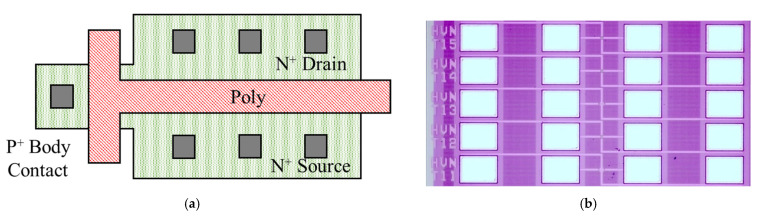
Layout of T-gate transistor used in our study and microscopic image: (**a**) T-gate transistor layout (the T-gate is used for body contact to suppress the floating-body effect); (**b**) microscopic image of the devices. The pads in the diagram are, from left to right, the gate, drain, source, and body.

**Figure 2 micromachines-13-01432-f002:**
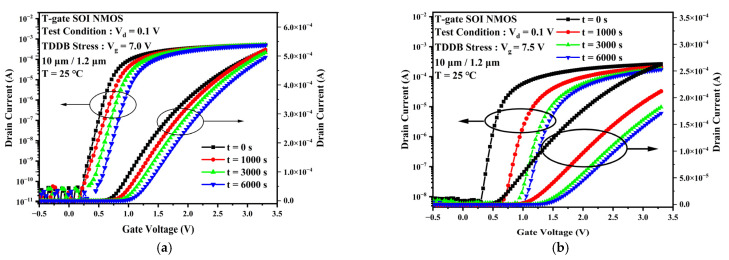
Degradation of the transfer characteristic curve of the device when the TDDB stress is applied (**a**) Vg = 7 V, (**b**) Vg = 7.5 V.

**Figure 3 micromachines-13-01432-f003:**
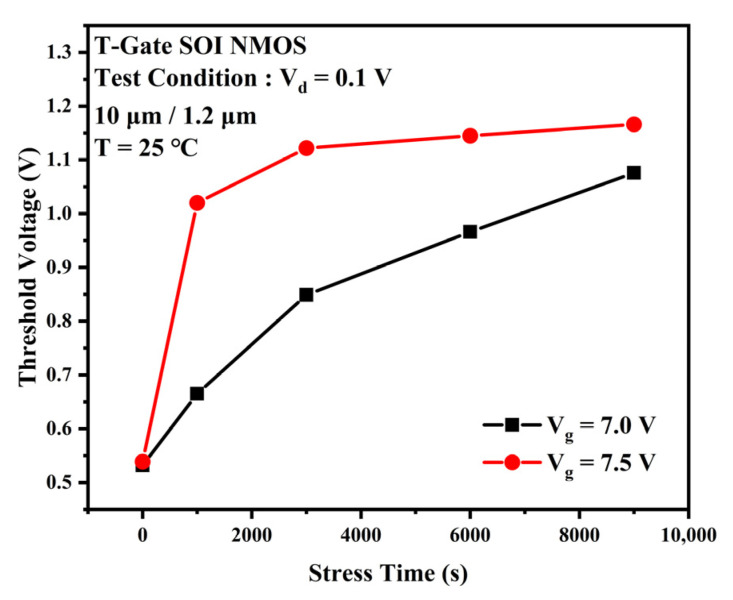
Degradation of the device threshold voltage when the TDDB stress is applied.

**Figure 4 micromachines-13-01432-f004:**
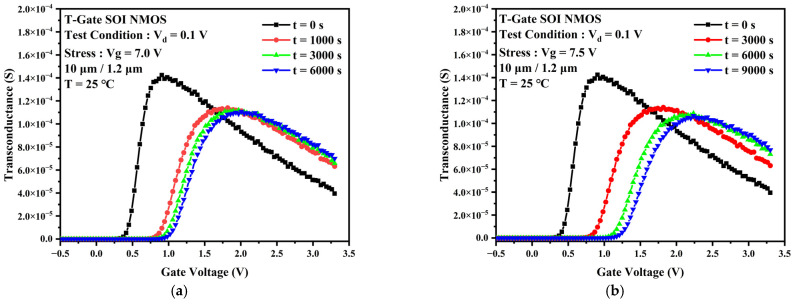
Degradation of transconductance of the device when the TDDB stress is applied: (**a**) Vg = 7 V, (**b**) Vg = 7.5 V.

**Figure 5 micromachines-13-01432-f005:**
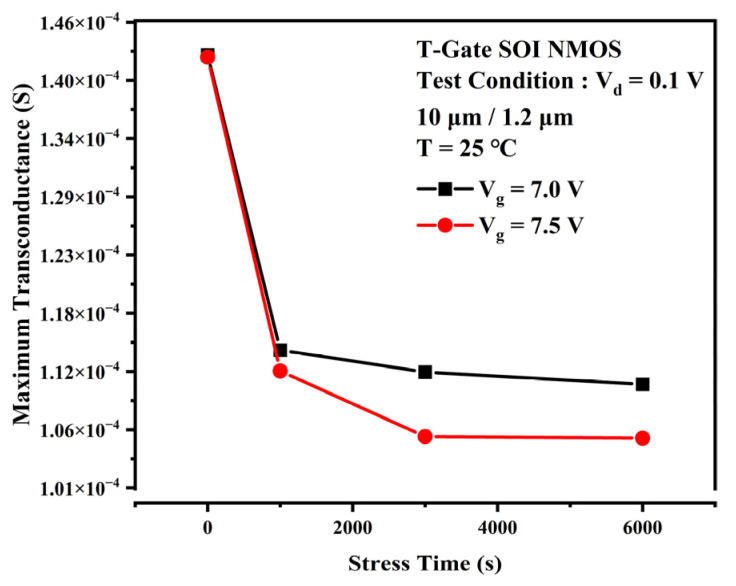
Degradation of the maximum value of device transconductance when TDDB stress is applied.

**Figure 6 micromachines-13-01432-f006:**
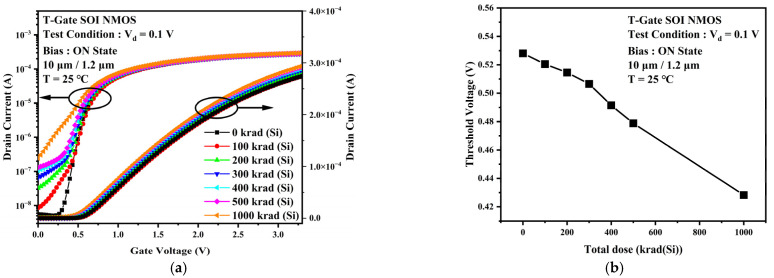
(**a**) Device transfer characteristic curves at different irradiation doses. (**b**) Threshold voltage at different irradiation doses.

**Figure 7 micromachines-13-01432-f007:**
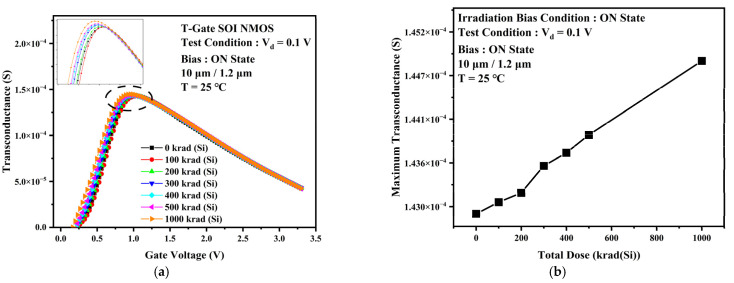
(**a**) Transconductance curves of devices at different irradiation doses. (**b**) Maximum transconductance values at different irradiation doses.

**Figure 8 micromachines-13-01432-f008:**
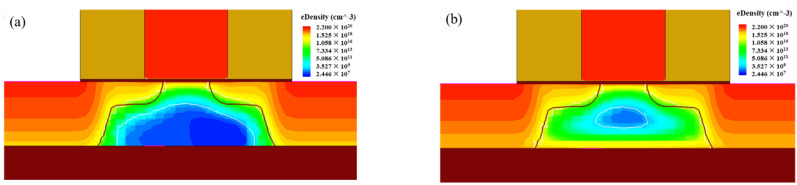
Simulation of channel electron number density in PDSOI device. (**a**) before irradiation, (**b**) after irradiation.

**Figure 9 micromachines-13-01432-f009:**
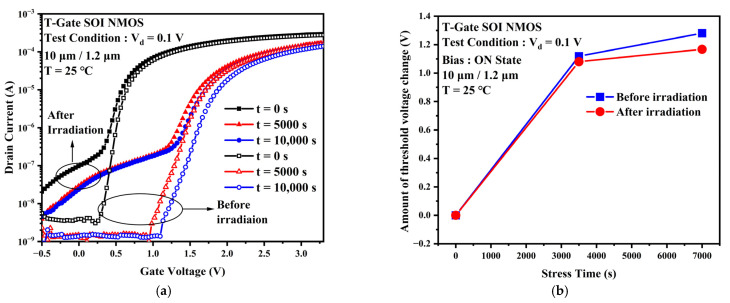
(**a**) Transfer characteristic curves before and after irradiation with TDDB stress applied. (**b**) Amount of threshold voltage changes when TDDB stress is applied before and after irradiation.

**Figure 10 micromachines-13-01432-f010:**
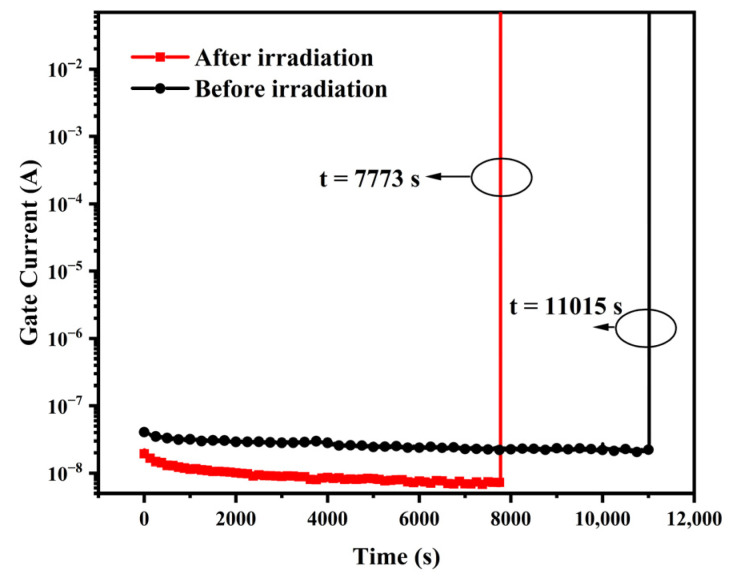
Comparison of TDDB breakdown application of sample devices before and after irradiation.

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
