# Peer review of "Investigation of the Combined Effect of Total Ionizing Dose and Time-Dependent Dielectric Breakdown on PDSOI Devices"

_micromachines, 2022, doi:10.3390/mi13091432_

Round 1

Reviewer 1 Report

I enjoyed the paper!

1. I have one important mandatory comment:

Please do not use the terms TID and TDDB in the title!

Write the names properly.

Furthermore, I have few questions which interests me:

2. Are you sure that the reported results will apply to all samples?

3. In addition: what is the source of the SOI wafers?

4. Where did you fabricate the devices?

Reviewer 2 Report

General comments:

The manuscript is good, however it is missing two things:

  1. An important and current reference within the context of the work;

      D. S. Monte, doi: 10.1109/I2MTC43012.2020.9129059

  1. Some considerations mainly comparing the behavior of the devices of this work with those of the reference above, including such comparisons should appear in the conclusions; just an example: the behavior of maximum transconductances after irradiation…, etc. 

Specific corrections:

  1. P1-L36 - What is STI?
  2. P1-L40 - Please, indicate the meaning of TDDB and PDSOI only the first time these terms appear in the Introduction. (i.e., here in L40).
  3. P1-L42 - Change to: … TDDB lifetime, whereas proton
  4. P2-L46 - Change to: … small, whereas the TDDB
  5. P3-L111 - Change to: There is
  6. P4 Fig2 - In my opinion there is something strange in Fig.2a on the right axis. I believe the zero point is actually 9∙10-5, so the curves wouldn't come together like it's shown there. Or would it be a "failure" of the equipment (semiconductor analyzer), which in its internal firmware makes a fit automatically!!! This issue needs to be clarified. The same must be observed for Fig.2b.
  7. P5-L152 - Change to: 3.2 Evaluation of the TID effect
  8. P6-L166 - The same problem as in Fig2 appears in Fig.6.
  9. P6-L168 - There is an error in the y-axis of Fig7b., or the axis has to have better resolution.
  10. P7-L193 - Correct the text punctuation.
  11. P7-L98 - Fig.8; Correct the legends: before irradiation; after irradiation.
  12. P7-L201…L202 - Improve the sentence: The amount … to irradiation.
  13. P8 Fig.9 - Show both curves on one graph only; Also Correct the legends, as in 11).
  14. P8-L242 - Change to: … parameters, whereas the TID …

Conclusion: 

Publish after the requirements and corrections have been made.

Reviewer 3 Report

This manuscript proposed an investigation of TID and TDDB for SOI device. This work is timely and will be useful to other researchers working in the area of radiation hardness technology. While the work is worthy of publication, the authors need to expand onto some details of the work described.

Please address the following comments:

1.      In Figure 2, is the length and width 10 um and 1.2 um? The scale mentioned in the title is 0.13um. Why does the author use the long channel device for investigation?

2.      Is there any material analysis data for discussion? Such as XPS, SIMS, XRD. Please try to add some material analysis data into the manuscript. It will be more convincing.

3.      The irradiation experiments were carried out by using a Co60 gamma irradiation. How about the uniformity? In order to further figure out the real mechanism, please try to measure TID and TDDB from at least 15 devices with different channel length or width.

4.      The on current is slightly improved after higher TID, and it seems that the mobility is slightly improved. What is the reason?

5.      English writing needs to be revised by a native English speaker.

Round 2

Reviewer 3 Report

1.      Please remove the description about 130 nm, if the length and width of device 10 um and 1.2 um, although that process is not advanced any more.

2.      The story proposed in this work is not persuasive enough, if there is not any material analysis data in the manuscript.

Round 3

Reviewer 3 Report

Please add the TCAD simulation data (Figure 1 in author's reply) into the manuscript, instead of the material analysis result.
